# Health Disparities and Climate Change: The Intersection of Three Disaster Events on Vulnerable Communities in Houston, Texas

**DOI:** 10.3390/ijerph19010035

**Published:** 2021-12-21

**Authors:** Omolola E. Adepoju, Daikwon Han, Minji Chae, Kendra L. Smith, Lauren Gilbert, Sumaita Choudhury, LeChauncy Woodard

**Affiliations:** 1Department of Health Systems and Population Health Sciences, University of Houston College of Medicine, 4849 Calhoun Road, Bldg 2, Houston, TX 77204, USA; lrgilbe2@central.uh.edu (L.G.); lwoodard@central.uh.edu (L.W.); 2Humana Integrated Health System Sciences Institute, University of Houston, Houston, TX 77204, USA; minji.chae@utexas.edu (M.C.); schoudh5@central.uh.edu (S.C.); 3Department of Epidemiology & Biostatistics, Texas A&M School of Public Health, College Station, TX 77845, USA; dhan@tamu.edu; 4Smith Research & Consulting LLC, Spring, TX 77386, USA; klsphd@gmail.com

**Keywords:** health disparities, environment, climate change, COVID-19, disasters

## Abstract

Although evidence suggests that successive climate disasters are on the rise, few studies have documented the disproportionate impacts on communities of color. Through the unique lens of successive disaster events (Hurricane Harvey and Winter Storm Uri) coupled with the COVID-19 pandemic, we assessed disaster exposure in minority communities in Harris County, Texas. A mixed methods approach employing qualitative and quantitative designs was used to examine the relationships between successive disasters (and the role of climate change), population geography, race, and health disparities-related outcomes. This study identified four communities in the greater Houston area with predominantly non-Hispanic African American residents. We used data chronicling the local community and environment to build base maps and conducted spatial analyses using Geographic Information System (GIS) mapping. We complemented these data with focus groups to assess participants’ experiences in disaster planning and recovery, as well as community resilience. Thematic analysis was used to identify key patterns. Across all four communities, we observed significant Hurricane Harvey flooding and significantly greater exposure to 10 of the 11 COVID-19 risk factors examined, compared to the rest of the county. Spatial analyses reveal higher disease burden, greater social vulnerability, and significantly higher community-level risk factors for both pandemics and disaster events in the four communities, compared to all other communities in Harris County. Two themes emerged from thematic data analysis: (1) Prior disaster exposure prepared minority populations in Harris County to better handle subsequent disaster suggesting enhanced disaster resilience, and (2) social connectedness was key to disaster resiliency. Long-standing disparities make people of color at greater risk for social vulnerability. Addressing climate change offers the potential to alleviate these health disparities.

## 1. Introduction

Over the past decade, Houston, Texas has become one of the most diverse cities in America [1], during which time it has also endured several federally declared disasters. Houston sits within Harris County, the nation’s fourth most populous county [2], and more than 60% of its residents identify as Hispanic or Black [3]. These minority populations are more than twice as likely to live below the poverty line as their White and Asian counterparts [4], despite living within 15 miles of higher-socioeconomic-status (SES) communities who enjoy 21 additional years of average life expectancy and $50,000 more in average income [5]. Researchers have documented how racial-minority communities experience multidimensional poverty and a lack of resources, including healthcare, transportation, healthy foods, and other basic needs [5]. Both historic and present governmental policies have created inequitable conditions for racial minorities [6], making the communities in which they live more susceptible to natural disasters. Given the increasing frequency of natural disasters precipitated by climate change, the importance of multi-hazard risk assessment is heightened. The city of Houston provides a unique lens with which to study this, as it has experienced successive disasters, including Hurricane Harvey and Winter Storm Uri, within the setting of the COVID-19 pandemic.

In recent years, drastic global climate change has contributed to multiple 500-year flood events that have devastated Houston communities. Hurricane Harvey was by far the most destructive, causing USD 125 billion in damages [7], and recovery efforts are still underway four years later. However, the storm caused more than physical damage for Houston-area communities with higher levels of poverty; these communities have experienced an increased prevalence of mental health symptoms consistent with post-traumatic stress disorder [8]. Additionally, one in six affected Texans have reported a new or worsening health condition due to Hurricane Harvey [9,10]. Minority residents with limited resources proved more likely to incur property damage or income loss and years afterward still struggle to recover from the natural disaster [9,10].

The disproportionate impact of COVID-19 on minority communities is well-documented. Hospitalizations hold the greatest disparity; rates are 4.7 times higher for non-Hispanic Blacks than for Whites (while deaths are 2.1 times higher and cases are 2.6 times higher, respectively) [11]. Furthermore, pre-existing health disparities affecting the African American community have only been exacerbated by factors including, but not limited to, job loss and food insecurity [12,13]. African American individuals living in Houston experience a higher burden of chronic disease and multidimensional economic, environmental, and social hurdles compared to their White counterparts [12,14]. These disparities, although amplified by COVID-19, are not new. Chronic diseases, along with the multidimensional economic and environmental hurdles, put African American communities in a vulnerable state to be equipped for severe climate change and a pandemic.

While reeling from Hurricane Harvey and the COVID-19 pandemic, Houston experienced yet another major emergency: the 2021 Winter Storm Uri. More than two-thirds of Texans lost electrical power during the February 2021 winter storm for an average of 42 hours, while almost half of Texans lost access to running water during this period [15]. Communities of color, as well as other historically marginalized communities, across the state of Texas were hit especially hard. African American communities of low SES were among the first to lose power and suffered through frigid and hard conditions due to poor infrastructure, such as lack of insulation [15]. Post-winter storm, Texas households faced large utilities bills due to line-loss charges which increased the price of electricity during the freeze. In fact, Harris County saw a 2800% increase in line-loss charges in February 2021 when compared to charges in the previous year [16]. Communities of color were less equipped to recover, making the disaster recovery rather difficult [17].

Although evidence suggests that these disasters disproportionately damage African American neighborhoods, resilience has played a major role in how African Americans bounce back [18,19]. The American Psychological Association (APA) defines resilience as “how easily individuals can adapt and recover from difficult situations including traumatic events, tragedy, serious health problems, and financial stressors” [20]. To depict the hypothesized relationships between successive disasters, population geography, race, and health disparities-related outcomes, we designed a framework diagram as shown in Figure 1. Given the undeniable link between climate change and increasingly damaging natural disasters [21], we consider the impact of climate change on resiliency in vulnerable communities. Communities made vulnerable due to geographic and racial factors experience greater burden of chronic disease, social vulnerability, and poverty, while also experiencing an increase in disaster resilience. 

Recognizing the negative trifecta effects of Hurricane Harvey, COVID-19, and winter storm Uri, we designed a mixed-methods study to examine the impact of successive disasters precipitated by climate change on resiliency in African American communities. Findings from this work can be used to develop policies to mitigate resulting inequities and increase preparedness for future climate change events. Multi-hazard risk assessments of this nature are significant because they provide a detailed understanding of hazard vulnerability interactions and can inform how specific geographic populations prepare for future disasters.

## 2. Methodology

*Quantitative methods:* Using the 2018 American Community Survey (ACS) data on Houston’s Super Neighborhoods [22], we identified four communities in the greater Houston area with prominent proportions of non-Hispanic African American (AA) residents: Kashmere Gardens (59.46%), Sunnyside (79.97%), Third Ward (64.57%), and Acres Home (55.11%). These four communities are historically well-established AA communities where the research team has previously worked. For example, Sunnyside, is one of the oldest AA communities located in south central Houston. Acres Homes is also a well-recognized residential area for primarily for African Americans in the northwestern area of Houston. We identified two more African American communities in the central Houston area in order to have more geographically diverse locations. As illustrated in Figure 2, these four communities were used in this study to represent the AA population of the entire study area. Data chronicling the local environment, including population demographics and community-level risk factors for pandemics and disaster events, were obtained from two primary data sources: the ACS and the Centers for Disease Control (CDC) Places project [23].

First, we used the CDC Places 2019 release to identify known risk factors of COVID-19 at a small-area neighborhood level, consistent with the current understanding of COVID-19 epidemiology [24,25]. These risk factors include demographics (population age over 65), population with chronic diseases (comorbidity factors), and health behavior-related factors (obesity and smoking) [25].

We subsequently used the 2014–2018 ACS five-year estimate to identify population and socio-structural characteristics of local communities that are important in determining the differential impact of disasters on at-risk communities [23]. A total of 10 small-area neighborhood variables were identified as potential community-level disaster-related risk factors in this study. These risk factors include demographic and socioeconomic status: population age over 65, population in poverty, unemployed, and per capita income), housing (crowded housing), education (no high school diploma), transportation (population without a vehicle), insurance coverage (population without health insurance coverage), and disability status. Previous research shows that higher educational attainment is linked to measures of increased disaster preparedness, perhaps because education level may be related to the degree to which individuals process information on risk-minimization [26]. 

These census tract level data were combined into a Geographic Information System (GIS) database to build a base map of the four AA communities and Harris County for further statistical and spatial analyses. GIS mapping methods were primarily used to depict spatial patterning of the COVID-19 and disaster risk factors between the target communities and other communities in study area. Descriptive statistics were employed to quantify the differences within and between the four target communities and other communities in Harris County. COVID-19 and disaster risk factors between the four AA communities and the rest of Harris County were compared, as were the demographic and socioeconomic characteristics of census tracts between the communities.

*Qualitative methods:* We conducted two focus groups with 26 community-dwelling African American adults between February–March 2021. The focus groups were conducted after Winter Storm Uri, which allowed us to assess the lived experiences of African American older adults during successive disaster events.

### 2.1. Qualitative Design and Eligibility

We conducted focus groups with participants who met the following criteria: (1) self-identified as an African American, (2) were at least 55 years of age, and (3) lived in Houston between 2018–2020. We recruited participants via e-mail and through Houston-based community partners working with the University of Houston College of Medicine. Interested participants responded to the recruitment invitation, and investigators scheduled them to engage in one of two focus group discussions, which were conducted virtually due to the COVID-19 pandemic. Participants received a USD 50 gift card for their participation. Study investigators with expertise in community health, health services research, and community-engaged research developed the focus group guide. The study was approved by the University of Houston Institutional Review Board (ID: STUDY00002584, approved on 7 October 2020).

### 2.2. Qualitative Procedures and Analysis

At the beginning of the focus groups, participants received information about the purpose of the study and provided informed consent indicating their agreement to participate. Focus groups were 90 minutes in duration and discussion centered on participant experiences in disaster planning and recovery, as well as community resilience in response to successive disasters. Participants shared their lived experiences through all three emergencies and disasters (Hurricane Harvey, COVID-19, and Winter Storm Uri).

We recorded and transcribed all focus group sessions. A note taker and facilitator took extensive notes during each session. We conducted a thematic analysis, with notes and recordings reviewed by two members of the research team with extensive experience in qualitative data analysis. The focus group facilitator was a member of the analysis team. The analysis team independently reviewed the notes and recordings to identify emerging themes for each domain. After independent coding concluded, the analysis team compared coding schemes to identify agreements and resolve disagreements. We conducted a second review to refine themes and identify anecdotes and quotes that were illustrative examples of each theme.

## 3. Results

Table 1 compares the community risk factors and other descriptive characteristics in the four selected communities compared to the rest of Harris County. There were statistically significant differences between the neighborhoods located within the four communities and those in the rest of the county for all COVID-19 risk factors, except for one (cancer prevalence). The four AA communities were older and had lower education levels, lower income levels, higher levels of poverty and unemployment, and greater disadvantage (as indicated by a higher percent population without a vehicle and a higher percent disabled population) than other Harris County residents. Summary statistics for additional variables, including COVID rates (incidence rate per 1000 and case fatality rates) and percent of homes flooded due to Hurricane Harvey, are also included in the table. The prevalence of coronary heart disease is 9.6% in Kashmere Gardens, 8.5% in Sunnyside, 6.1% in Third Ward, and 7.6% in Acres Home. These rates are equal to or greater than the state-wide prevalence of 6.1% [27]. Additionally, the burden of diabetes is high, affecting 22.5% of the population in Kashmere Gardens, 20.1% in Sunnyside, 14.9% in Third Ward, and 17.5% in Acres Home. These rates are up to two times higher than the state-wide diabetes burden of 11.4% [28,29]. Even higher is the prevalence of obesity: 47.5% of Kashmere Gardens residents, 46.0% of residents in Sunnyside, 39.6% in Third Ward, and 42.6% in Acres Home have obesity. Compared to the Texas obesity prevalence of 34.8% in 2018, all four neighborhoods exhibit a substantially greater burden of the disease [21]. The standardized mortality ratios (SMR) for Kashmere Gardens, Sunnyside, Third Ward, and Acres Home are 1.6, 1.5, 1.2, and 1.3, respectively. In other words, each of these four African American communities observe more deaths among their residents than expected.

Because we observed significantly greater exposure to the majority of community risk factors examined, Figure 3, Figure 4 and Figure 5 depict the spatial patterning of key health disparity factors for the four African American communities compared to all other communities in the county. These maps illustrate spatially varying patterns of disease burden using diabetes as an example (Figure 3), social vulnerability (Figure 4), and poverty (Figure 5) within and outside the AA communities.

Two themes emerged from the thematic data analysis:

*Finding 1: Previous disasters, including Hurricane Harvey and the COVID-19 pandemic, prepared minority populations in Harris County to better handle subsequent disasters, suggesting enhanced disaster resilience.* Participants noted that previous disasters provided important coping and preparation skills, although each occurrence was a major stress. While all participants were left without water and power for days during Winter Storm Uri’s extremely low temperatures, many reported additional issues following the restoration of power, including burst pipes and limited water supplies. Participants highlighted the parallels between the natural disasters of hurricanes and the pandemic with the unprecedented winter storm, focusing on how hurricane preparations were very similar to the preparations for the winter storm.

“*(Hurricane) Harvey prepared me better. Harvey prepared me mentally and physically better (for the winter storm)*.”(Participant L).

Common disaster preparation practices included having bottled water and food on hand and filling a bathtub with water to have a water source for flushing toilets.

“*Harvey prepared us because we were already in disaster mode and how do we take care of this and how do we take care of our families and make sure we are prepared next time. In the ice storm, as soon as they made the weather reports and projects, it was like ‘make sure these things are done’—fill tank, fill tub just like a hurricane. I had charged all devices, and I charged the chargers, and I already had candles and I have flashlights. Same prep for a hurricane you did for the ice storm*.”(Participant C).

Another participant reported that she purchased additional supplies when the winter storm was being predicted.

“*We have to stay prepared. It could happen tomorrow or in a couple of months, we just have to stay prepared.” (Participant D). Another participant further emphasized this point. “Disasters and emergencies can be anything, it doesn’t have to be a pandemic or ice storm. Just naturally always being prepared*.”(Participant B).

“*We know now that if something is coming, not to wait until the last minute, not to wait until everybody is in the store grabbing things off the shelves. As soon as you hear of something you, you go and make your preparations*.”(Participant C).

Overall, several participants shared a realization about the need for disaster preparedness as they applied lessons from prior disaster events to weather the winter storm. Although armed with coping skills from previous disasters, they all reported that each occurrence was still stressful, but somewhat easier to manage than the previous.

*Finding 2: Social connectedness was key to disaster resiliency; previous disasters reinforced the importance of staying connected to family and friends during the winter storm.* Our findings suggest that the preparation plans and concerns surrounding disasters expanded beyond an individualistic view. Participants underscored the importance of checking in on other people, including loved ones and neighbors, during the recent disaster events.

“*We need to check on the elders more. There were people that have been found since the thaw, who were just dead at home and people were not aware of that. If someone had said ‘we haven’t heard from him and the power is out’ but then nobody went to check and the person was already frozen to death (…). That was during this (winter storm) but even during COVID, I think we need to check (on each other) more frequently because anything can happen and we shouldn’t go days (without) talking to them*.”(Participant L).

This collective approach to check in on each other was rooted in taking care of others who may need help.

“*We need to start talking about things that we can do, and think about others all the time, always thinking about how we can better help others survive this. Because we may be doing ok, but somebody else may not*.”(Participant L).

Several participants noted that because they were prepared for the winter storm, they were able to assist family and friends who needed help.

“*Because I was prepared in the sense that I could help my son who was in his apartment without power for four days*.”(Participant P).

“*When (Hurricane) Harvey came, I got lucky and bought a generator. (…) So I had a generator when this thing (the winter storm) happened so that helped us tremendously. (…) I charged my next-door neighbor’s battery with the generator*.”(Participant C).

These comments reflect respondents’ perceptions on the need to remain connected during disasters to ensure the health and safety of their social networks. Additionally, they demonstrate the role of relationships and echo the importance of integrating a community lens into future disaster planning, response, and recovery efforts.

## 4. Discussion

This study examined the impact of successive climate change disasters coupled with the COVID-19 pandemic on African American communities in Houston. To the best of our knowledge, this is the first study to triangulate successive disasters and how such events can further exacerbate health disparities, specifically in an urban setting. The use of a methodology that incorporates both qualitative and quantitative research data in a single study allows the research question to be thoroughly studied from different perspectives. Importantly, this study raises awareness on the intersection between climate change fueled disaster events and health disparities. Our findings suggest that long-existing racial disparities are exacerbated during these successive disasters. Particularly for Black residents in the greater Houston area, who already bear a disproportionate share of the COVID-19 burden and may still be recovering from the catastrophic hit of Hurricane Harvey in 2017, Winter Storm Uri was the third major strike in recent years that has knocked many off their feet [30].

Signals of climate change and its blatantly destructive character offer an undeniable need for concern. Though patterns of climate change have been both predicted and observed since the early 19th century, the topic still remains one of the most debated and resisted points in the current political atmosphere. Our findings suggest that the observable and tangible products of climate change are present and harmful in Houston, a region that is known for its vibrant diversity. The impact of climate change is undisputable through the cycle of 100-year storms, hurricanes, and city-wide flooding. The area has more flood-related casualties and property loss than any other locality in the USA [31], stemming from the coastal placement of the region combined with suboptimal prevention policies and flood mitigation planning. While Hurricane Harvey affected all demographics, socially vulnerable populations like racial minorities and low-income residents were particularly hit the hardest and last to recover from such a catastrophic event. Of the four communities we examined, Kashmere Gardens was most affected by Hurricane Harvey with 82% of homes flooded during the natural disaster, while 37% of homes in Sunnyside, 23% of homes in Third Ward, and 32% of homes in Acres Home were flooded. Though the storm landed in Texas on 25 August 2017, its impact on residents continues over three years later as Texans recover and rebuild from what they lost from the natural disaster. According to the 2020 survey from the Hobby School of Public Affairs at the University of Houston, nearly 20% of survey participants who were displaced by Hurricane Harvey are still in temporary housing [30,32]. These four communities of color are likely still rebuilding and recovering from the storm at a slower rate than other predominantly White communities [32].

Although climate-induced disasters may have unbiased origins, their impacts are greater among poor, minority, elderly, and immigrant populations. In the recent example of the 2021 state-wide freeze, Houston, and Texas as a whole, bore witness to a large fracture in the distributive framework for basic resources, such as electricity, food, and water. The historic Winter Storm Uri left over four million people out of power in Texas and the racial minority groups living in marginalized neighborhoods were the first ones to experience power outages [33]. African American communities in neighborhoods with poor housing were disproportionately affected by the storm and took the biggest hit. These communities already lacked resources and had dilapidating infrastructure, having been neglected for years. While coping strategies relied heavily on prior disaster response strategies, our findings also highlight the importance that minority communities place on social connectivity to get through difficult times.

## 5. Conclusions

In this study, we found greater social vulnerability and risks in minority communities. Work by Chae et al. (2020) suggests that Houston residents may experience these vulnerabilities differently not only because of race but also because of where they reside [34]. As geographic residence interacts with structural differences in power and social biases, the tangible products are lower life expectancy [35], high uninsured rates [36], and poorer physical and mental health among racial/ethnic groups. The presence and magnitude of these inequities will not fade as events caused by climate change grow in frequency and intensity. Rather, they will continue to work in tangent with the current frameworks of power that incite structural marginalization.

Throughout these successive disaster events, community support networks, including charities, governmental organizations, hospitals, and faith-based organizations, played key roles in alleviating the negative health and wellbeing impacts on the populations studied. For example, several faith-based organizations partnered with food pantries to host food drives that provide healthy nutritious options to residents. Community-based organizations raised funds to help with home repairs, busted pipes, and water damage, while others assisted residents with disaster assistance applications. Despite these efforts, the findings from this work suggest that the response to successive disaster events remains in its infancy and the need for culturally sensitive approaches to address successive disasters in unique racial/ethnic groups remains unaddressed.

As the US begins to reestablish itself, with efforts such as the rejoining of the Paris Climate Agreement and the Green New Deal proposal, our nation, as a whole, must also come to understand the gravity and time sensitivity that the climate change crisis holds and its salient capability to impact healthcare on a comprehensive scale. Stemming the negative impacts of climate change has the potential to alleviate health disparities.

## Figures and Tables

**Figure 1 ijerph-19-00035-f001:**
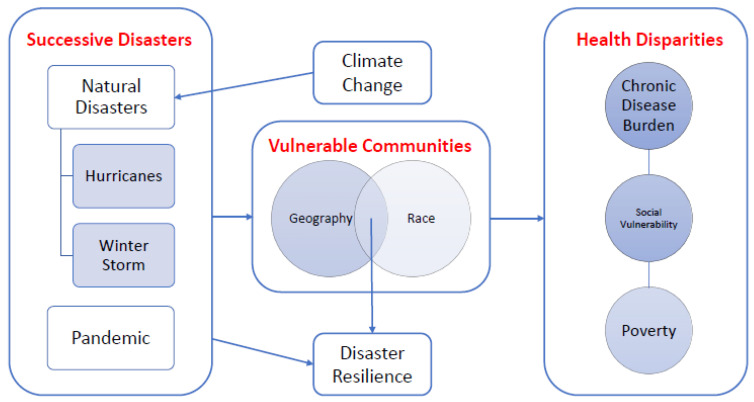
Conceptual Framework depicting the hypothesized relationships between successive disasters (and role of climate change), population geography and race, and health disparities related outcomes.

**Figure 2 ijerph-19-00035-f002:**
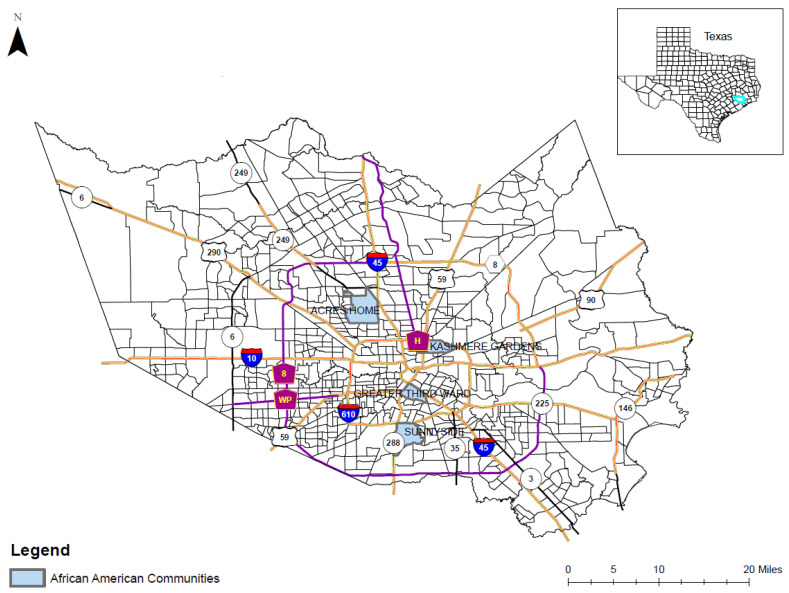
Location of four predominantly African American population communities located within Harris County, TX.

**Figure 3 ijerph-19-00035-f003:**
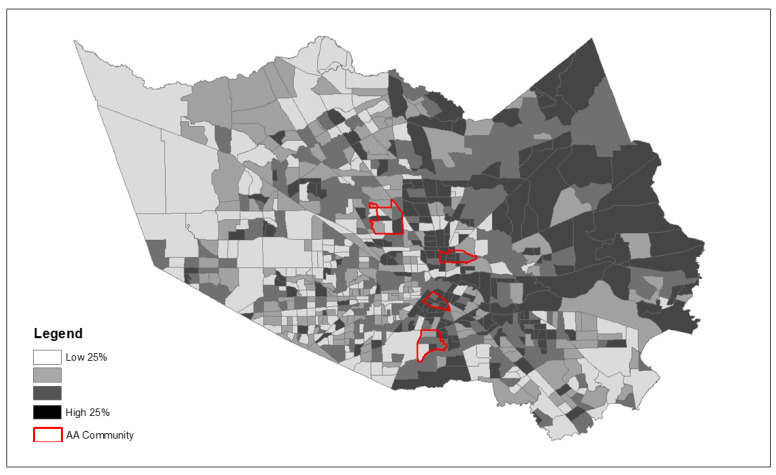
Spatial patterns of community risk factors for disease burden—Diabetes Prevalence (%) (Four AA communities indicated by red border on top of Harris County census tracts).

**Figure 4 ijerph-19-00035-f004:**
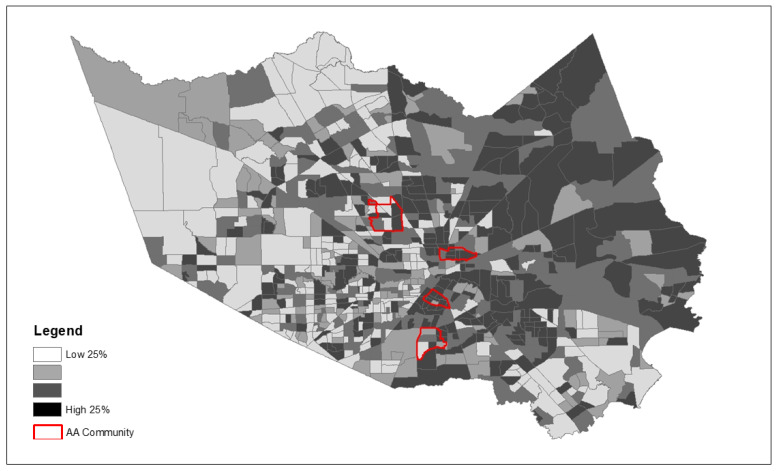
Spatial patterns of community risk factors—Social Vulnerability Index (Four AA communities indicated by red border on top of Harris County census tracts).

**Figure 5 ijerph-19-00035-f005:**
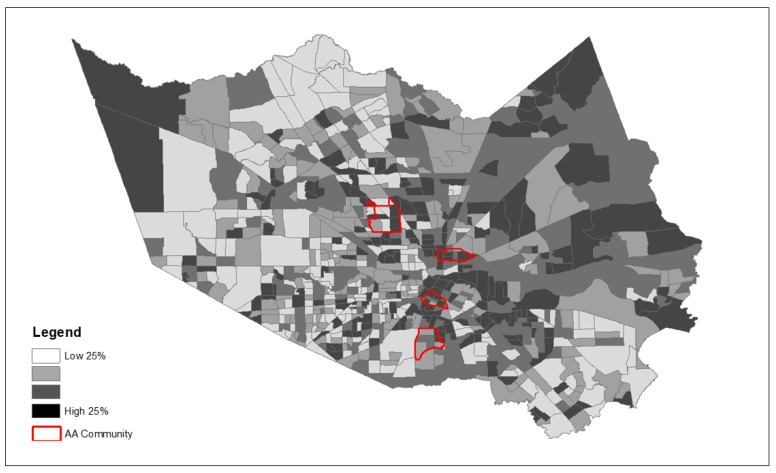
Spatial patterns of community risk factors for disaster—Population in Poverty (%) (Four AA communities indicated by red border on top of Harris County census tracts).

**Table 1 ijerph-19-00035-t001:** Community Risk Factors for Disaster Vulnerability in 4 African American Communities vs. Harris County, TX.

	Harris County		Kashmere Gardens		Sunnyside		Acres Home		Third Ward	
	Mean	SD	Mean	SD	Mean	SD	Mean	SD	Mean	SD
**COVID-19 Risk Factors**										
Population Age > 65 (%)	12.1 *	5.2	13.3	3.1	13.4	5.3	11.5	4.5	10.5	6.8
Smoking (%)	20.2 **	4.9	23.0	1.4	21.7	5.3	19.7	4.7	16.9	5.9
Obesity (%)	43.7 **	6.2	47.5	1.7	46.0	6.9	42.6	6.4	39.5	9.3
Hypertension (%)	40.3 **	5.5	44.8	4.4	43.7	9.6	38.5	6.4	35.0	12.4
Diabetes (%)	18.7 **	4.0	22.5	1.7	20.1	5.8	17.5	4.4	14.9	7.3
Cancer (%)	5.3	1.4	5.5	0.5	5.5	1.2	5.5	1.0	4.5	1.6
Stroke (%)	5.4 **	1.1	6.5	0.9	6.2	2.1	4.8	1.4	4.2	1.6
Coronary Heart Disease (%)	7.9 **	1.7	9.6	0.8	8.5	2.6	7.6	1.4	6.1	3.0
Chronic Obstructive Pulmonary Disease (%)	8.1 **	1.8	9.7	0.9	8.9	2.9	7.8	1.7	6.2	3.0
Asthma (%)	11.0 **	1.2	11.4	0.8	11.8	1.8	10.5	1.6	10.6	2.0
Chronic Kidney Disease (%)	4.2 **	0.8	4.9	0.4	4.6	1.3	3.9	0.8	3.4	1.5
Non-Hispanic African American (%)	12.1 *		79.5		80.0		65.1		64.6	
Standardized Mortality Ratio (SMR)	20.2 **	0.3	1.6		1.5		1.3		1.2	
**Disaster Risk Factors**										
Below Poverty (%)	27.8 **	13.6	29.8	7.1	31.2	14.3	27.9	17.2	23.5	13.8
Unemployed (%)	10.2 **	6.8	9.5	4.0	15.2	9.8	10.5	5.2	6.6	4,0
No High School Diploma (%)	22.8	12.0	33.7	7.5	21.4	10.7	12.4	9.7	24.5	10.3
Disabled (%)	15.3 **	7.0	20.1	6.9	18.4	8.4	12	5.8	12.1	3.0
Crowded Housing (%)	5.4	3.6	6.7	4.2	4.6	2.9	3.0	2.8	6.9	3.1
No Vehicle (%)	15.4 **	10.9	14.8	6.0	16	12.7	21.4	14.9	16.9	6.7
No Health Insurance (%)	31.6	8.4	38.8	4.9	31.2	6.8	24.7	6.9	32.1	8.7
Per capita income ($)	22,235 **	14,298	16,276		18,752		27,810		24,930	
Median Household Income ($)			30,149		25,240		40,442		27,860	
Homes Flooded during Harvey (%)	22		82		37		32		23	
COVID Incidence (per 1000)	54.5	19.7	50–74		50–74		50–74		<25	
COVID Case Fatality Rate (%)	0.9	0.7	1–1.5		1–1.5		1–1.5		>1.5	

* *p* < 0.05; ** *p* < 0.01.

## Data Availability

Qualitative data is based on focus group discussion and per disclosure provided to participants, we are not able to share individual level information.

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
