# Peer review of "Health Disparities and Climate Change: The Intersection of Three Disaster Events on Vulnerable Communities in Houston, Texas"

_ijerph, 2021, doi:10.3390/ijerph19010035_

Round 1

Reviewer 1 Report

The authors have improved on the presentation of data  and briefly mention the conceptual frameworks for investigation. Various social networks and their influence are not mentioned in the introduction. The conclusion fails to highlight the role of these social networks in multi hazard vulnerability/resilience. This needs to be addressed.

The conceptual framework of multi-hazard risks has been mentioned and the data presents the multiple hazards in the area of study.

Author Response

The authors have improved on the presentation of data  and briefly mention the conceptual frameworks for investigation. Various social networks and their influence are not mentioned in the introduction. The conclusion fails to highlight the role of these social networks in multi hazard vulnerability/resilience. This needs to be addressed. The conceptual framework of multi-hazard risks has been mentioned and the data presents the multiple hazards in the area of study.

Response:

We thank the review for this comment. We have now added the following to the introduction. We are optimistic that this will provide additional context on the role of social networks and their influences:

“Aligning with the social network theory, community, voluntary and not-for-profit organizations also play an important role in supporting and building resilience, as evidenced by the city of Houston’s influential networks of charities, healthcare systems, community- and faith-based institutions. The Houston’s Mayor Office launched the “Resilient Houston Strategy” in February 2020 to focus on improving resiliency at every scale, including expanding knowledge regarding emergency preparedness, provide risk awareness programs in Houston for any emergency, and provide disaster recovery resources and training to Houston residents to build resilience and reduce vulnerability (“Resilient Houston,” 2020), (Chu & Yang, 2020). Volunteers from faith-based organizations and non-profit organizations organized food drives and served families with food and clean water since the pandemic and the winter storm.

We have also addressed this theme in the conclusion, as shown below:

“Throughout these successive disaster events, community support networks, including charities, governmental organizations, hospitals, and faith-based organizations, played key roles in alleviating the negative health and wellbeing impacts on the populations studied. For example, several faith-based organizations partnered with food pantries to host food drives that provide healthy nutritious options to residents. Other community-based organizations raised funds to help with home repairs, busted pipes, and water damage, while others assisted residents with disaster assistance applications. Despite these efforts, the findings from this work suggests that the response to successive disaster events remains in its infancy and the need for culturally sensitive approaches to address successive disasters in unique racial/ethnic groups remains unaddressed.”

 We have also now included a conceptual framework of the relationships between successive disasters (and role of climate change), population geography and race, and the hypothesized relationship with health outcomes and social vulnerability.

Reviewer 2 Report

This study seeks to evaluate resilience and exacerbation of health disparities among African American populations residing in Harris County, TX, with attention to three succeeding weather events – Hurricane Harvey, Winter Storm Uri, and COVID-19. This is a laudable goal as such populations will increasingly be prone to compounding disaster due to climate change and are already at a disadvantage with respect to other population groups. I like the mixed methods approach and I think this is an important area of study. However, there are some issues that I recommend addressing before publication.

General comments:

  • There are some spelling and grammar errors throughout the paper; please do a thorough read-through to address these
  • The tense switches from past to present and back in the space of a few sentences, e.g. in the abstract and Intro; it’s typical to use past tense when discussing work already completed. Please make it consistent one way or the other.

Abstract & Intro

First – the paper seems to be lacking a cohesive story outside the population and the location. The abstract first talks about disaster exposure, then resilience, then 11 COVID-19 risk factors that are not defined. What is the overarching framework? Define this and stick with it. A conceptual diagram could help.

The Intro begins with a discussion of Latinos and African Americans, which leads the reader to believe the paper will be about both populations until the paper starts talking exclusively about the latter population. It’s sufficient to discuss disparities between African Americans and others in Harris County; including the Latino information is misleading and unnecessary.

The last sentence in the first paragraph speaks of “Hurricane Harvey…within the setting of the COVID-19 pandemic,” but Harvey occurred more than two years prior to the onset of the pandemic in the US, so this is misleading. How do the authors conceptualize the relationships between these time-variant disasters and the outcomes of interest?

¶4 – “While there is mixed evidence…it is undeniable that these marginalized communities…” This sentence contradicts itself. If there is not sufficient evidence, then the language should be tempered.

¶4 – “spent a disproportionate amount of income on their utilities” – please cite evidence to support this.

Methodology

Why these four communities? Presumably there are others with high proportions of African American residents that were not selected. What was the process for identifying these neighborhoods and selecting them for analysis?

Each of the variables in the analysis should be clearly defined, and we should know from the Intro what their proposed relationship to “health disparities” or “resilience” is. This is lacking in the Intro and it’s not totally clear what the outcome is – “health disparities?” Which ones?

Why is HS diploma noted as “communication barrier?” Where’s the evidence supporting this?

There should be a quantitative analysis subheading under which the quant analysis is clearly explained. After reading this and looking at the results, it seems to me that descriptive mapping is the only thing that occurred, i.e. no “spatial analysis” was actually performed, or at least it wasn’t included in the manuscript.

Either way, there should be a clear explanation of all analytical steps such that a reader could duplicate the process.

One of the eligibility criteria was that the respondent was “living in Houston.” Were there any steps taken to ensure that all participants had resided in Houston continuously since before Harvey? Otherwise, how would they assess the impact of all three disasters?

What was the amount of the gift card?

Results

The maps are very difficult to read due to the scaling, the amount of information included, and the similarity between the neighborhood outline and the background color. There’s also no indication as to the units of geography being displayed (census tracts?).

I suggest only retaining the relevant info for each map (e.g. is SVI necessary to include – if so, this should also be theorized and defined earlier in the paper), and either zooming in on the 4 areas, or creating insets for each so that we can more clearly see the patterns.

Using ColorBrewer is a good way to ensure divergent color schemes.

Why the four outcomes shown in Fig. 3? What is their relationship to the story being told here? Why these disparities over others? How do these fit with the 11 risk factors mentioned in the abstract?

Same questions for Fig. 4. These two figures are especially difficult to see as well.

The Intro and Discussion suggest the qual and quant results are “triangulated” to provide a better picture of disaster in these communities, but I don’t see any triangulation. The quotes included from the focus groups have little to do with the results shown in the maps. Triangulation involves telling the same story with the quant and qual data but from different methodological perspectives. This seems to tell two disparate stories. Again, an overarching framework / theoretical model / organizing philosophy would really help.

Discussion

The discussion draws back out to focus on how this study tells us that minoritized populations are harder hit by disaster, but this is not new and not really a contribution to the literature. What is more interesting and less covered is a mixed methods approach to these issues and assessment of disasters as successive and compounding. I suggest the authors give their paper a framing that speaks more to these unique aspects of their work and less to what is already known on the subject.

Author Response

General comments:

  • There are some spelling and grammar errors throughout the paper; please do a thorough read-through to address these
  • The tense switches from past to present and back in the space of a few sentences, e.g. in the abstract and Intro; it’s typical to use past tense when discussing work already completed. Please make it consistent one way or the other.

Response: We thank the reviewer for this comment. We have revised the manuscript to ensure the tense use is consistent

Abstract & Intro

First – the paper seems to be lacking a cohesive story outside the population and the location. The abstract first talks about disaster exposure, then resilience, then 11 COVID-19 risk factors that are not defined. What is the overarching framework? Define this and stick with it. A conceptual diagram could help.

Response: We thank the reviewer for this comment and agree that it strengthens the concepts we explore in this manuscript. We have now included a conceptual framework of the relationships between successive disasters (and role of climate change), population geography and race, and the hypothesized relationship with health outcomes and social vulnerability.

The Intro begins with a discussion of Latinos and African Americans, which leads the reader to believe the paper will be about both populations until the paper starts talking exclusively about the latter population. It’s sufficient to discuss disparities between African Americans and others in Harris County; including the Latino information is misleading and unnecessary.

Response: We thank the reviewer for this comment and have now edited to include the African American population that this study focuses on.

The last sentence in the first paragraph speaks of “Hurricane Harvey…within the setting of the COVID-19 pandemic,” but Harvey occurred more than two years prior to the onset of the pandemic in the US, so this is misleading. How do the authors conceptualize the relationships between these time-variant disasters and the outcomes of interest?

Response: We thank the reviewer for this comment. Hurricane Harvey and the flooding happened in August-September 2018, 18 months before the pandemic. However, recovery efforts were still underway and several vulnerable communities are still recovering. You can still see remnants of the Hurricane (and its’ floods) destruction in Houston. We conceptualize these time-variant disasters from the lens of climate change and successive disasters and how those interact with race and geography. Our inclusion of the conceptual framework makes the case stronger. 

¶4 – “While there is mixed evidence…it is undeniable that these marginalized communities…” This sentence contradicts itself. If there is not sufficient evidence, then the language should be tempered.

Response: We appreciate the reviewer for this comment. The quoted section from the Introduction has now been changed to: “Communities of color, as well as other historically marginalized communities, across the state of Texas were hit especially hard.”

¶4 – “spent a disproportionate amount of income on their utilities” – please cite evidence to support this.

Response: The sentence containing the quoted section has now been changed to: “Post-winter storm, Texas households faced large utilities bills due to line-loss charges with the increased price of electricity during the freeze. In fact, Harris County saw a 2800% increase in line-loss charges in February 2021 when compared to charges in the previous year. 11” We have cited an article from the Houston Chronicle for this addition.

Methodology

Why these four communities? Presumably there are others with high proportions of African American residents that were not selected. What was the process for identifying these neighborhoods and selecting them for analysis?

Response: We have provided justification by including the following in the methods:

“Using the 2018 American Census Survey (ACS) data on Houston’s Super Neighborhoods, we identified four communities in the greater Houston area with prominent proportions of non-Hispanic African American residents: Kashmere Gardens (59.46%), Sunnyside (79.97%), Third Ward (64.57%), and Acres Home (55.11%), according to”. Of all Houston Super Neighborhoods, these four communities are included in the top 10 Super Neighborhoods with the highest proportions of African Americans.  In addition. these four communities are historically well-established AA communities where the research team has previously worked. For example, Sunnyside, is one of the oldest AA community located in south central Houston. Acres Homes is also well-recognized residential area for primarily for African Americans, in north western area of Houston. We identified two more African American communities in the central area in order to have more geographically diverse locations. As illustrated in Figure 1, these four communities were used in this study to represent AA population of the entire study area.”

Each of the variables in the analysis should be clearly defined, and we should know from the Intro what their proposed relationship to “health disparities” or “resilience” is. This is lacking in the Intro and it’s not totally clear what the outcome is – “health disparities?” Which ones?

Response: With the inclusion of a conceptual framework (again we thank the reviewer for this suggestion), we have clearly included a description of the variables examined, and the hypothesized relationships. Our work is an initial step in examining the relationships between successive disasters (and role of climate change), population geography and race, and health disparities outcomes. We used maps to highlight 3 health disparities-related outcomes—disease burden, using diabetes as an example, social vulnerability and poverty.

Why is HS diploma noted as “communication barrier?” Where’s the evidence supporting this?

Response: Previous research shows that higher educational attainment is linked to measures of increased disaster preparedness, perhaps because education level may be related to the degree individuals may process information on risk-minimization (Menard et al., 2011). Therefore, the present study denotes not having a high school diploma as a communication barrier to disaster preparedness or resilience and further, a disaster-related risk factor.

Menard LA, Slater RO, & Flaitz J. (2011). Disaster preparedness and educational attainment. Journal of Emergency Management, 9(4): 45-52. DOI: 10.5055/jem.2011.0066

There should be a quantitative analysis subheading under which the quant analysis is clearly explained. After reading this and looking at the results, it seems to me that descriptive mapping is the only thing that occurred, i.e. no “spatial analysis” was actually performed, or at least it wasn’t included in the manuscript.

Either way, there should be a clear explanation of all analytical steps such that a reader could duplicate the process.

Response: We have completely revised the spatial analysis to align with the new conceptual framework. In this new analysis, we only use 3 maps to show the health disparities-related outcomes: disease burden, using diabetes as an example, social vulnerability and poverty. We also use clearer maps, and the analytic steps are much easier to follow for readers who would like to replicate the process

One of the eligibility criteria was that the respondent was “living in Houston.” Were there any steps taken to ensure that all participants had resided in Houston continuously since before Harvey? Otherwise, how would they assess the impact of all three disasters?

Response: As part of recruiting for the focus groups, we ensured all participants lived in Houston between 2018-2020. This is now included in the manuscript.

What was the amount of the gift card?

Response: A $25 gift card was provided to program participants. This is now included in the manuscript.

Results

The maps are very difficult to read due to the scaling, the amount of information included, and the similarity between the neighborhood outline and the background color. There’s also no indication as to the units of geography being displayed (census tracts?).

I suggest only retaining the relevant info for each map (e.g. is SVI necessary to include – if so, this should also be theorized and defined earlier in the paper), and either zooming in on the 4 areas, or creating insets for each so that we can more clearly see the patterns. Using ColorBrewer is a good way to ensure divergent color schemes. Why the four outcomes shown in Fig. 3? What is their relationship to the story being told here? Why these disparities over others?Same questions for Fig. 4. These two figures are especially difficult to see as well.

Response: As advised, we trimmed the maps significantly and only retained relevant maps, in alignment with our conceptual model. recognizign we had too many variables, we completely revised the spatial analysis to align with the new conceptual framework. In this new analysis, we only use 3 maps to show the health disparities-related outcomes: disease burden, using diabetes as an example, social vulnerability and poverty. We also use clearer maps, and the analytic steps are much easier to follow for readers who would like to replicate the process

The Intro and Discussion suggest the qual and quant results are “triangulated” to provide a better picture of disaster in these communities, but I don’t see any triangulation. The quotes included from the focus groups have little to do with the results shown in the maps. Triangulation involves telling the same story with the quant and qual data but from different methodological perspectives. This seems to tell two disparate stories. Again, an overarching framework / theoretical model / organizing philosophy would really help.

Response: We thank the reviewer for this comment. As suggested, we have now included a conceptual framework. The results from the focus group underscore the impacts of Hurricane Harvey, the pandemic and Winter storm Uri. The maps show differential effect of Hurricane Harvey floods and the pandemic on African American communities vs. the rest of Houston. This is what we refer to as triangulation of 3 events from both quantitative and qualitative methods. This work focuses on vulnerability and the intersection of three Disaster Events on Vulnerable Communities. Through the unique lens of successive disaster events (Hurricane Harvey and Winter Storm Uri) coupled with the COVID-19 pandemic, we assessed disaster exposure on minority communities in Harris County, Texas. Further, we examine the implications for mitigating post-exposure health consequences in at-risk communities and preparing for future climate change events.

Discussion

The discussion draws back out to focus on how this study tells us that minoritized populations are harder hit by disaster, but this is not new and not really a contribution to the literature. What is more interesting and less covered is a mixed methods approach to these issues and assessment of disasters as successive and compounding. I suggest the authors give their paper a framing that speaks more to these unique aspects of their work and less to what is already known on the subject.

Response: We agree that the use of a mixed methods approach for the assessment of disasters as successive and compounding is interesting. We provide additional implications for this in the conclusions, as shown below:

“In this multi hazard risk assessment study, we found greater social vulnerability and risks in minority communities. Work by Chae et al (2020) suggests that Houston residents may experience these vulnerabilities differently not only because of race, but also because of where they reside. As geographic residence interacts with structural differences in power and social biases, the tangible products are lower life expectancy, high uninsured rates, and poorer physical and mental health, among racial/ethnic groups. The presence and magnitude of these inequities will not fade as events caused by climate change grow in frequency, rather they will continue to work in tangent with the current frameworks of power that incite structural marginalization.

Throughout these successive disaster events, community support networks, including charities, governmental organizations, hospitals, and faith-based organizations, played key roles in alleviating the negative health and wellbeing impacts on the populations studied. For example, several faith-based organizations partnered with food pantries to host food drives that provide healthy nutritious options to residents. Other community-based organizations raised funds to help with home repairs, busted pipes, and water damage, while others assisted residents with disaster assistance applications. Despite these efforts, the findings from this work suggests that the response to successive disaster events remains in its infancy and the need for culturally sensitive approaches to address successive disasters in unique racial/ethnic groups remains unaddressed.

As America begins to reestablish itself, with efforts such as the rejoining of the Paris Climate Agreement and the Green New Deal proposal, our nation, as a whole, must also come to understand the gravity and time sensitivity that this crisis holds and its salient capability to impact healthcare on a comprehensive scale. Stemming the negative impacts of climate change has the potential to alleviate health disparity gaps.”

Round 2

Reviewer 2 Report

Thank you for the opportunity to review this manuscript a second time. While I think the current version represents a substantial improvement over the initial submission, I still think there are problems that need to be addressed prior to acceptance. 

First, the manuscript still has grammar errors and should be reviewed for mistakes. 

The third paragraph in the Intro makes a number of claims, none of which are cited. Every claim requires at least one citation.

Second, while I appreciate the inclusion of a theoretical framework, it is not well integrated into the paper and the concept is unclear. The framework should not be in the Methods but in the Intro, and it should be integrated with the lit review. There should also be a textual explanation of the framework, including how it ties all facets of the paper together. 

It's very unclear how resilience fits with health disparities, or why there's a double arrow to one box and a single arrow from another. What exactly is this meant to convey? 

Further, there's no inclusion of the actual chronic disease conditions that are reported in the results. Where do these fit in?

Finally, is climate change necessary to include in the diagram? Disasters occur with or without it, so this also needs to be more clearly integrated.

The first sentence of the Quant Methods paragraph is incomplete, and citations should be included for both the Super Neighborhoods reference and the CDC Places project. 

It's not sufficient to explain why HS education is an appropriate measure for "communication barrier" in the response to review; it needs to be explained in the manuscript itself. 

The maps are much clearer now, thank you. However, the neighborhood labels are quite hard to read in Figure 2. It would help to give them a halo or otherwise format the text for legibility.

Lastly, I still don't understand how the chronic disease prevalence rates relate to the rest of the paper. There is no mention of chronic illness in the Intro and they aren't included in the conceptual diagram. It doesn't make sense to present "results" that have not been theorized or otherwise outlined in the paper. There needs to be a clear reason for presenting these specific conditions as results that tell us something about climate change, resilience and disadvantage, and the case has not been made. 

Author Response

Response to Reviewer (Round 2)

Thank you for the opportunity to review this manuscript a second time. While I think the current version represents a substantial improvement over the initial submission, I still think there are problems that need to be addressed prior to acceptance. 

Response: We thank the reviewer for their continued suggestions to improve the manuscript.

First, the manuscript still has grammar errors and should be reviewed for mistakes. 

Response: We have reviewed the manuscript for grammatical errors and mistakes.

The third paragraph in the Intro makes a number of claims, none of which are cited. Every claim requires at least one citation.

Response: We have reviewed the third paragraph of the introduction and have added citations for the claims made. In total, there are some 9 additional references.

Second, while I appreciate the inclusion of a theoretical framework, it is not well integrated into the paper and the concept is unclear. The framework should not be in the Methods but in the Intro, and it should be integrated with the lit review. There should also be a textual explanation of the framework, including how it ties all facets of the paper together. 

Response: We appreciate the reviewer’s suggestion and have integrated the theoretical framework, as well as a textual explanation, into the introduction.

It's very unclear how resilience fits with health disparities, or why there's a double arrow to one box and a single arrow from another. What exactly is this meant to convey? 

Response: This double arrow is meant to convey the complex relationship between resilience and disaster events: experiencing disasters can improve community resilience to disasters, but also resilience itself impacts community experience of disaster events. However, we understand the confusion from the double arrows, so we have replaced it with a single arrow.

Further, there's no inclusion of the actual chronic disease conditions that are reported in the results. Where do these fit in?

Response: The chronic disease burden fits in the framework through “Disease Burden.” We understand that this may be confusing, so we have updated the framework to indicate “Chronic Disease Burden” instead of “Disease Burden” as we previously showed.

Finally, is climate change necessary to include in the diagram? Disasters occur with or without it, so this also needs to be more clearly integrated.

Response: We agree with the reviewer – disasters do occur with or without climate change. However, climate change causes greater intensity of natural disasters, and therefore, with the pressing issue of climate change, we must consider the impact it will have on vulnerable communities. We expand on this idea in the discussion and conclusion of the manuscript.

The first sentence of the Quant Methods paragraph is incomplete, and citations should be included for both the Super Neighborhoods reference and the CDC Places project. 

Response: We have reviewed the grammatical errors in the mentioned paragraph and have included the mentioned citations. They are respectively references #22 and #23.

It's not sufficient to explain why HS education is an appropriate measure for "communication barrier" in the response to review; it needs to be explained in the manuscript itself. 

Response: We thank the reviewer for their suggestion in improving clarity of the manuscript. We have included an explanation under the Quantitative Methods section of the manuscript for why not a high school diploma having is considered an appropriate measure of a communication barrier.

The maps are much clearer now, thank you. However, the neighborhood labels are quite hard to read in Figure 2. It would help to give them a halo or otherwise format the text for legibility.

Response: We have made edits to the map, to improve readability.

Lastly, I still don't understand how the chronic disease prevalence rates relate to the rest of the paper. There is no mention of chronic illness in the Intro and they aren't included in the conceptual diagram. It doesn't make sense to present "results" that have not been theorized or otherwise outlined in the paper. There needs to be a clear reason for presenting these specific conditions as results that tell us something about climate change, resilience and disadvantage, and the case has not been made. 

Response: We do mention chronic disease in the introduction (lines 80-86), and we have changed the conceptual diagram to indicate “Chronic Disease Burden” (instead of “Disease Burden,” as it was before). We also expand on the complex relationships between chronic disease burden, geography, race, and disaster events. Our manuscript emphasizes that vulnerable communities that experience heavy chronic disease burden, social vulnerability, and poverty are also the same communities that are vulnerable in the setting of disaster events. We hope that the changes we’ve made make this message clearer.

Round 3

Reviewer 2 Report

I have no additional comments at this time.

This manuscript is a resubmission of an earlier submission. The following is a list of the peer review reports and author responses from that submission.

Round 1

Reviewer 1 Report

This article brings to fore the discussions on the disproportionate impacts of, and responses to successive multiple hazards. In addition, the focus on socio- economic factors presents a critical dimension to disaster relief planning. 

The publication may  benefit from the following:

  • To what extent did environmental factors such as location of the study areas in the path of the hurricane, landscape etc exacerbate the impacts?
  • How influential were community, voluntary or non-profit organisations in supporting and building resilience? Mapping community networks might be helpful in identifying the role of such in building resilience-see social network theory.
  • Though the authors refer to quantitative and qualitative methods, there seems to be minimal reference to the theoretical aspects of multi hazard risk assessment underpinning the study and research design. These aspects needs to be addressed in the introduction section.
  • More concrete conclusion s needed.

Reviewer 2 Report

  1. The introduction has a long preamble, which is in many useful, but it also pushes what passes for a statement of purpose of the paper to the end of the introduction, almost two pages into an otherwise short paper of 15 pages. Also, the statement of purpose should be more tightly written, ideally with a testable hypothesis. “examining” barely qualifies as a purpose; its more of a task. The paper is on an important topic, so, make the writing more compelling and enticing.
  2. First paragraph of the methods section (please use page number in future version of the manuscript to make the review work possible!): where do these data come from? Year?
  3. It seems you focused on the four neighborhoods mentioned in para. Of the methods section. Why these 4? Why not neighborhoods with large Latinx population instead or as well?
  4. What do acronyms KS, LW, and OA mean?
  5. Table 1. Reports on the covid-19 risk factors for all 4 communities together. It would be useful to report this for each individually, and also report how much variability the is within each of the four communities, between neighborhoods.
  6. There is mention of covid-19 risks factors. The factors retained are based on national views. The point here is to know whether this really translates in higher infection and death rates in each of these communities. By now the data should be available publicly. I am asking for authors to provide this info to validate the risk factors in comparison to other types of communities to make the argument much tighter.
  7. Figure 1 should not be in the results section but in the section where the communities are introduced. Same for Table 1. These are not results.
  8. Maps are of poor quality. Please provide some of the broader context , like where we are in Tx (this is an international journal!), add a scale and compass, as well as a legend.
  9. Authors present a lot of data in the form of tables and maps, for the four targeted communities, as well and all other communities in Houston. They then proceed to describe this information. Given the richness of the date on hand, I see no reason for not conducting a rigorous statistical analysis of these data, with covid infection and death rates as dependent variables, and others as independent variables. A multivariate analysis is ESSENTIAL to avoid the pitfalls of collinearity.
  10. Findings 1 and 2 is largely conjectural and require to be substantiated through careful and rigorous statistical analysis.
  11. The paper needs to fully substantiate many assertions that are made. The connection with resilience developed as a results of exposure to other unexpected events is interesting, but this is rather anecdotal at the paper stands at this time.

Round 2

Reviewer 2 Report

Thanks for responding to my comments on the original version of the manuscript.  I have carefully read the revised manuscript and the responses authors provided to the original comments.

I do note once again that authors have not added page numbers (as requested!), which continues to make the work of any manuscript reviewer much more time consuming than it should be.

Thanks for you providing some of the context for the conduct of this study. This does help to put things in perspective.

I find the revisions made to be quite light, and far from what I expected. In fact, authors are rather dismissive of my point of view.

The responses provided make it clear this work is very preliminary, rushed, with very little focus on covid, but rather on vulnerability in general. The writing remains rather unsubstantiated.

I continue to see Table 1 as providing background information, as a starting point for the research. It is not part of the research per se, and therefore should NOT be in the results section.

While authors speak of their research as using a mixed methods approach, I certainly some qualitative research, but no real analysis, as this is essentially limited to doing a few interviews, with do not seem to have been in-depth, and without real attempt to tie to the social, employment, education, economic and health history of the person, family or community. Furthermore, to be mixed, a methodology needs to have something quantitative to “mix” the qualitative with. The only “quantitative” information provided here is from the ACS in the form of tables and a few maps. There is nothing that is specific to the Covid-19 pandemic. Yes, this would be useful, yes this would be essential. My very point, is that would love to see good mixed methods research on this topic. Here however, the authors leave it all for future work. I did not learn from this manuscript anything I didn’t already know about vulnerability. Nothing that did not make it on the media over the past 9 months. I am very disappointed authors have nothing new to share with readers that would shed light on this critical important aspect of the pandemic.

I cannot recommend anything other than a rejection at this stage.